# Automatic Refractive Error Estimation Using Deep Learning-Based Analysis of Red Reflex Images

**DOI:** 10.3390/diagnostics13172810

**Published:** 2023-08-30

**Authors:** Glenn Linde, Renoh Chalakkal, Lydia Zhou, Joanna Lou Huang, Ben O’Keeffe, Dhaivat Shah, Scott Davidson, Sheng Chiong Hong

**Affiliations:** 1oDocs Eye Care, Dunedin 9013, New Zealand; 2University of Sydney, Sydney, NSW 2050, Australia; 3Choithram Netralaya, Indore 453112, India; 4Dargaville Medical Centre, Dargaville 0310, New Zealand; 5Public Health Unit, Dunedin Hospital, Te Whatu Ora Southern, Dunedin 9016, New Zealand

**Keywords:** refractive error, myopia, fundus imaging, red reflex

## Abstract

**Purpose**: We evaluate how a deep learning model can be applied to extract refractive error metrics from pupillary red reflex images taken by a low-cost handheld fundus camera. This could potentially provide a rapid and economical vision-screening method, allowing for early intervention to prevent myopic progression and reduce the socioeconomic burden associated with vision impairment in the later stages of life. **Methods**: Infrared and color images of pupillary crescents were extracted from eccentric photorefraction images of participants from Choithram Hospital in India and Dargaville Medical Center in New Zealand. The pre-processed images were then used to train different convolutional neural networks to predict refractive error in terms of spherical power and cylindrical power metrics. **Results**: The best-performing trained model achieved an overall accuracy of 75% for predicting spherical power using infrared images and a multiclass classifier. **Conclusions**: Even though the model’s performance is not superior, the proposed method showed good usability of using red reflex images in estimating refractive error. Such an approach has never been experimented with before and can help guide researchers, especially when the future of eye care is moving towards highly portable and smartphone-based devices.

## 1. Introduction

The most common cause of distance vision impairment is an uncorrected refractive error, of which myopia is the leading disorder [1]. The global prevalence of myopia and high myopia (≤−5.00 D) is rising rapidly and they are expected to affect 50% (4.7 billion) and 10% (1 billion) of the global population by 2050, respectively [2]. Myopia is an emergent public health issue recognized by the World Health Organization (WHO) as one of the leading causes of preventable blindness [3].

High myopia is associated with significant ocular co-morbidities, many of which cause irreversible vision loss, including myopic macular degeneration, macular neovascularization, glaucoma, and retinal detachment [4,5,6]. Even low myopes (−1.00 to −3.00 D) experience a two-fold increase in developing myopia-associated ocular morbidities [7,8]. Without effective intervention, vision loss from myopia-associated ocular pathology is expected to increase sevenfold [2].

Myopia and associated complications cause a significant individual public burden with an estimated annual cost of USD 202 billion worldwide [9]. The socioeconomic burden of myopia is expected to be further exacerbated by the declining age of onset and faster progression in children [10,11,12]. Effective intervention to control myopic progression is critical to reducing the disease burden for individuals and wider society [12]. To address the growing myopic burden, there is an urgent need for a cost-effective myopia screening program to provide efficacious treatment.

Vision screening programs, such as the New Zealand B4 School Check (B4SC) and Australian Statewide Eyesight Screening Program (StEPS), rely on visual acuity testing to detect refractive error [13]. Visual acuity-based screening programs are screener-dependent and produce high false positives with poor positive predictive values [14,15]. Autorefractor-based screening, although more straightforward and cost-effective, is limited by its necessity for maintaining position and visual fixation for a sufficient and substantial amount of time, which can be difficult in uncooperative children [16].

### 1.1. Red Reflex Test

The red reflex test, also known as the Brückner test, was pioneered by Brückner in 1962 [17] and has been postulated as a potential screening test to detect refractive errors. The test is performed in a darkened room using a coaxial light source, typically a direct ophthalmoscope, with the examiner positioned at an arm’s length distance from the subject. The examiner then looks through the ophthalmoscope and focuses on both corneas simultaneously, noting a red reflex in each pupil [18]. A normal test consists of symmetrically bright and round red reflexes in both eyes [19]. Asymmetry of the reflexes may be associated with strabismus, anisometropia, media opacities or cataracts, retinoblastoma, colobomas, and pigment abnormalities of the posterior pole [18,20]. The location and size of the pupillary crescent can also indicate the presence of ametropia [21,22,23]. Kothari [20] used the pupillary crescent from the red reflex test to determine and classify a patient’s refractive state as myopic (inferior crescent > 1 mm), hyperopic (superior crescent ≥ 2 mm), or astigmatic (crescent at any location decentred by >1 clock hour). The test was determined to have a sensitivity of 91% and specificity of 72.8%, with overall data supporting the use of the Brückner test in screening for refractive errors. A recent study based on smartphone photography reported by Srivastava et al. [24] has used Brückner’s reflex and has demonstrated reliability in identifying amblyogenic conditions in school children.

### 1.2. Photoscreeners

Several studies [20,23,25,26] have also looked into photo screening, a technique that works on the principle of the Brückner test. Photorefraction devices capture images of the light crescent generated on the pupillary red reflex. Results have been comparable to the Brückner test [23,27]. However, unlike the Brückner test, which relies on a live, moving subject, photoscreener images permit an unlimited amount of time for interpretation [20]. Images can be either manually graded or automatically graded by prediction algorithms [28]. A wide variety of photoscreeners are now available, including the PlusoptiX, Welch Allyn Spot Vision Screener, and the smartphone-based GoCheckKids app [29,30,31].

### 1.3. Artificial Intelligence in Ophthalmology

In the past few years, there has been a tremendous expansion of research into artificial intelligence (AI) applications in health care [32]. Much of the progress in AI research is achieved through the use of deep learning (DL), a subset of AI that utilizes convolutional neural networks (CNNs), comprised multiple layers of algorithms, to perform high-level feature extraction [33,34]. DL allows a machine to automatically process and learn from raw datasets and analyze complex non-linear relationships [35,36]. One of the key benefits of using DL algorithms in medicine has been in medical imaging analysis and screening.

In ophthalmology, DL systems have been applied in the classification of retinal fundus images, visual field results, and optical coherence tomography (OCT), to aid in the detection of ocular pathologies such as refractive error [16,37], diabetic retinopathy [38,39,40], glaucoma [41,42], retinopathy of prematurity [43], macular oedema [44], and age-related macular degeneration [42,45,46,47]. DL has also been shown to predict clinical parameters such as high coronary artery calcium scores by extracting the hidden features of retinal fundus images, which human clinicians may not have picked up [48,49].

Varadarajan et al. [37] developed a deep learning-based prediction model with high accuracy in estimating refractive error from retinal fundus images. More recently, Chun et al. [16] trained a CNN with eccentric photorefraction photographs to categorize refractive errors into mild, moderate, and high myopia or hyperopia. Although the deep learning model demonstrated an overall accuracy of 81.6% in estimating refractive error compared to cycloplegic examination [16], the generalizability of the results is limited unless the same smartphone model and camera settings can be replicated.

This paper proposes an experimental and novel method to estimate refractive error using color and infrared (IR) red reflex images acquired by an economical and portable smartphone-based fundus camera device known as nun IR [50].

## 2. Methods

### 2.1. Data Preparation

The proposed study was a pooled analysis of two prospective cohort studies conducted at Dargaville Medical Center in Northland, New Zealand, and Choithram Hospital in India.

The study at Dargaville Medical Center aimed to evaluate the usability of retinal images taken by a smartphone-based fundus camera in a rural general practice setting. Patients aged 16 years and older who presented to Dargaville Medical Center with visual disturbance, headache, hypertensive urgency, transient ischaemic attack, or stroke were invited to participate in the study, which took place over one year from 15 November 2021 to 23 November 2022. Patients who could not consent, were non-ambulatory, in need of resuscitation, or deemed too unwell by the triage nurse or attending physician were excluded. Written informed consent was obtained from all 152 participants.

At Choithram Hospital, 360 patients aged 20–60 years were recruited. Exclusion criteria included any condition or disease that could potentially affect photorefraction and the ability to obtain a reliable photo image, such as media opacity (corneal scarring, cataracts), posterior segment pathology (retinal detachment, age-related macular degeneration, diabetic retinopathy, vitreous haemorrhage), nystagmus, neurological impairment, and poor cooperation. All patients provided written informed consent before participation.

Two types of red reflex images, IR and color, were captured from each patient using the nun IR fundus camera. An optometrist took photographs obtained from Choithram Hospital, while a clinician captured those from Dargaville Medical Center. Refractive error metrics, including spherical and cylindrical power, were obtained using an autorefractor.

Images were captured in a dark room with a nun IR fundus camera attached to an Android smartphone, as shown in Figure 1. The patient was instructed to look straight ahead at a green fixation light in the distance and not blink. This was to reduce accommodation, where changes in focus can dynamically affect the shape of the lens. Thus, if accommodation is not effectively controlled, this would potentially alter the refractive power of the lens and, consequently, the accuracy of the red reflex images.

Two images were taken for each patient: one of the right eye at a distance of 8cm and one of the left at a distance of 8 cm. Each image was then saved as a full-color red reflex image and an IR red reflex image, giving each patient a total of four red reflex images. The images were then cropped to include only the pupil, as shown in Figure 2.

### 2.2. IR-Based Imaging with nun IR

The nun IR handheld fundus camera is non-mydriatic, using IR light to allow the retina to be viewed without causing pupillary constriction. It sends out paraxial rays into the subject’s eye as shown in Figure 3.

### 2.3. Observation of Crescents

When viewing the anterior segment of the eye with the nun IR, two crescents typically appear (Figure 2). The observed crescents are far apart in patients with myopia, while in those with hyperopia, they are merged. A clear differentiation is seen when examining a model eye (Heine Ophthalmoscope Trainer) with the nun IR, as shown in Figure 4.

In the model eye, it was noted that when the crescents are far apart, the spherical power is negative, and when the crescents overlap, the spherical power is more positive. This phenomenon was also observed in actual patients. The top row of Figure 5 shows two visible crescents far apart in a myopic eye (−8.00 D). The second row of Figure 5 shows crescents merged in an emmetropic eye (0.00 D). These figures indicate that the positions of crescents in red reflex images could help predict myopia.

### 2.4. Grading Classification with CNN

After data preparation, the DL-based AI platform MedicMind [51] (Figure 6) was used to train a grading classifier to predict spherical and cylindrical power.

MedicMind uses the open-source framework TensorFlow with the CNNs Inception-v3 (Figure 7) and EfficientNet (Figure 8). Inception-v3 and EfficientNet are pre-trained models, although EfficientNet has a more advanced architecture. We implemented the transfer learning technique in both models to improve training accuracy and speed. MedicMind resized images to 299 × 299 for Inception-v3 and 384 × 384 for EfficientNet, as these are the required sizes for the pre-trained models. MedicMind also applied data augmentation to the red reflex images, changing the contrast, aspect ratio, flipping, and brightness to reduce overfitting. The last classification layer of each CNN was modified to become a grading classifier with one output, trained with Euclidean distance as the loss function, as opposed to softmax. The RMSProp optimizer was used for back-propagation.

### 2.5. Training, Validation, and Evaluation

We used the data from Choithram Netralaya as the training set. There were 357 patients from Choithram Hospital who were primarily of Indian ethnicity. The training set included 288 IR and 528 color images, as some photographs were excluded due to poor quality. The Choithram data had a median spherical power of −2.00 D, meaning that the patients were largely myopic. Accuracy was measured using the red reflex images of 152 patients from Dargaville Medical Centre as a validation set, which included 143 IR images and color images. Photographs that needed to be of better quality were excluded. Autorefraction results were provided as the ground truth. Thus, approximately 80% of images were used for training and 20% for validation, which were then evaluated with the MedicMind DL model to predict refractive error versus ground truth. All training data were scaled so that the median was 0.5, and predictions were considered correct if the predicted and the ground truth values were over 0.5 or if the predicted and ground truth values were less than 0.5.

## 3. Results

### 3.1. CNN Grading Results

The EfficientNet grading classifier gave 63% accuracy for spherical power when training with IR images and 70% for cylindrical power. For color red reflex images, 64% accuracy was obtained for spherical power and 57% for the cylinder.

The Inception-v3 grading classifier was of consistently lower accuracy than EfficientNet, with 55% accuracy for spherical power and 49% for cylinders with IR images. For color red reflex images, Inception-v3 achieved 54% accuracy for spherical power and 50% for the cylinder (Table 1).

For the grading of IR red reflex images with EfficientNet, 84% sensitivity and 47% specificity were obtained, being more pronounced in color with 94% sensitivity and 29% specificity. The grading classifier is good at predicting true positives but not as good at predicting true negatives.

Results were validated with the dataset from Dargaville Medical Centre. This dataset included predominantly patients of Māori (indigenous people of New Zealand) and European ethnicity. The data were cropped and processed using the same technique as Choithram, leaving 143 IR images. The distribution of spherical power is shown in Figure 9.

Figure 9 shows that patients were more hyperopic in the Dargaville medical center dataset (median spherical power of +0.50 D) compared to the Choithram hospital dataset (median spherical power of −2.00 D).

An accuracy of 52% was obtained when inference was performed on the 143 images from the Dargaville dataset using the DL models trained with the Choithram dataset. The training was also directly performed on the Dargaville dataset, giving 52% accuracy.

### 3.2. Categorizing into Crescent Types

The accuracy of both grading classifiers was consistently 70% or lower. A higher accuracy is necessary to be useful for myopia screening. IR images were manually categorized based on crescent types to determine how accuracy could be improved, as shown in Figure 10.

The spherical power of each image in the dataset was divided into crescent-type categories. Figure 11 shows that images belonging to category A, with crescents furthest apart, are universally myopic (negative spherical power), but those in category D can be either myopic or non-myopic. This is also observed with high sensitivity for both IR and color images (84% and 94%, respectively) and low specificity (47% and 29%, respectively), showing that the DL model can predict myopia considerably accurately but is not as accurate in detecting non-myopia.

Spherical power overlap can be seen in each category. The degree of overlap is more pronounced for cylinder, where there appears to be no correlation between crescent type and cylinder power, as shown in Figure 12.

Power overlap between different crescent types may be why the grading classifier does not give a higher accuracy.

### 3.3. Multiclass Classification with CNN

A multiclass classifier was trialed to improve the grading classifier results. The IR images were put into two bins, depending on whether the spherical power was greater or less than the median spherical power of −2.00 D. A multiclass classifier was then trained on the two classes.



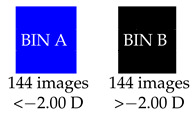



The following graph (Figure 13) shows the distribution of spherical powers for the 288 IR images:

After training a multiclass classifier for both Inception-v3 and EfficientNet, we obtained 70% accuracy for spherical power and 55% for the cylinder with EfficientNet for IR images, and 64% accuracy for spherical power and 60% for the cylinder with EfficientNet for color images (Table 2). Inception-v3 was consistently of lower accuracy for both IR and color images. Overall, color red reflex images gave lower accuracy than IR red reflex images when training with a multiclass classifier.

To improve the accuracy of the multiclass classifier, we trialed removing a certain percentage of images bordering on either side of the median so that the difference in spherical power between the two bins was more pronounced. Figure 14 shows the spherical power distribution when the middle 30% of images from the median was removed.

By removing 20% of the IR images bordering on either side of the median, the accuracy of a multiclass classifier was approximately 75% for spherical power and 50% for cylinder, with similar results for both EfficientNet and Inception-v3. After removing 20% of the color images bordering on either side of the median, accuracy was lower in terms of spherical power and cylinder for both EfficientNet and Inception-v3.

The removal of 1%, 5%, 10%, 20%, 30%, and 40% of images was trialed using Inception-v3 spherical power values, with the removal of 20% giving the best results (Table 3). However, removing the images only marginally improved accuracy.

### 3.4. Increasing Contrast

Increasing the contrast of images was also trialed to enhance the visibility of the crescents (Figure 15). This was tried for spherical power with EfficientNet with an accuracy of 69%. This was lower than the accuracy of 75% that was obtained before increasing contrast, indicating that increased contrast does not help improve accuracy.

### 3.5. Exclusion of Small Images

For EfficientNet, we trialed removing IR images that were small in size and less than 180 pixels wide. This resulted in a lower accuracy of 68% with the EfficientNet multiclass classifier, compared to the previous 74% (Table 4). The lower accuracy may have been due to fewer numbers (80 images) in each bin.

We also tried removing small-sized color red reflex images that were less than 450 pixels wide, as color images are much wider than IR images. We then trained spherical power with an EfficientNet multiclass classifier. We obtained 60% accuracy compared to 68% before removal. These results suggest that removing smaller images from the color red reflex images also does not improve accuracy.

### 3.6. Combining IR and Color

Rather than training on IR and color red reflex images separately, each IR and color image was placed side by side and combined into one picture in the hope that it would improve overall accuracy (Figure 16). The left side of the picture contained the IR image, and the right side contained the color image. Again, this did not improve accuracy, giving similar results to when trained separately (Table 5).

### 3.7. Summary of Results

The red reflex technique helps predict myopia with high accuracy. When crescents are type A (far apart), they are universally myopic, but if they are type D (merged), they can be either myopic or non-myopic (Figure 17).

The red reflex images above are all type A, where the crescents are apart and all myopic. This is also indicated by the high sensitivity values for predicting myopia with a grading classifier (84% and 94% for both IR and color).

However, when the red reflex images are all type D, only the image on the left is not myopic, with the other two images being myopic (Figure 18). This is also demonstrated in the low specificity values for IR and color image-based systems (47% and 29%, respectively).

The difficulty in classifying type D crescents as myopic or non-myopic was verified by the evaluation of the Dargaville dataset against Choithram-trained models. In the Dargaville dataset, there were far fewer type A images and more type D (median spherical power of +0.50 vs. −2.00 for Choithram). The low number of type A IR images and the high number of type D IR images in the Dargaville dataset is consistent with the low accuracy of 52% obtained with training the Dargaville dataset. Figure 19 shows a graphical summary of various training techniques used for the EfficientNet-based classifier and their performances.

## 4. Discussion

This study experimented with the possibility of using an AI-based model to predict refractive error from red reflex images taken using a smartphone-based portable fundus camera. The successful implementation of such a model would allow myopic individuals at risk of progression to be triaged and monitored remotely and digitally. This could reduce the number of clinical visits and improve treatment compliance, convenience, and patient satisfaction, thereby translating to saved time and costs for healthcare providers and consumers.

An effective screening test should be precise, economical, efficient, portable, and simple to administer. Although photoscreeners have gained considerable traction in research for their potential use in large-scale vision screening, the significant costs associated with many of these devices limit their practicality, particularly in developing and low-income countries. Nowadays, smartphones represent an integral part of modern life. Mobile applications among healthcare professionals in routine clinical practice are also becoming increasingly widespread, allowing direct access to medical knowledge, journals, calculators, and diagnostic tools [52]. As smartphone usage is so ubiquitous, photographs can be easily obtained virtually from anywhere in the world, overcoming the geographical and financial barriers of many photoscreeners.

Several studies have investigated the accuracy of smartphones for photorefractive screening [30,53,54]. Gupta et al. [53] utilized smartphone photography to screen for amblyogenic factors and found that all high refractive errors ≥ 5 D were successfully detected, although moderate refractive errors (3.25–5.00 D) revealed false negatives. However, the authors excluded low refractive errors due to the red reflex appearing normal, and the total number of subjects with a moderate or high refractive error was only 22, thereby limiting the applicability of their results. Another study [54] assessed the detection of amblyogenic risk factors using the GoCheckKids smartphone app. As both of these studies [53,54] evaluated the generalized prediction of amblyogenic risk factors and did not look at refractive error specifically, we could not directly compare the performance of our method to theirs. Arnold et al. [30] used the GoCheckKids app to identify refractive amblyopia risk factors, reporting a sensitivity of 76% and a specificity of 85%. However, these figures were achieved with manual grading of images by trained ophthalmic imaging specialists, which would involve substantial costs if this method were used for mass vision screening. An AI-based screening system could alleviate the burden of limited human resources, particularly in developing countries, and help economize this process.

In recent years, there has been a surge in the research and development of DL models to predict refractive error. In 2018, Varadarajan et al. [37] showed that contrary to existing beliefs, DL models could be trained to predict refractive error from retinal fundus images. Similarly, DL algorithms by Tan et al. [45] outperformed six human expert graders in detecting myopic macular degeneration and high myopia from retinal photographs. Another novel study [55] applied a DL algorithm to estimate uncorrected refractive error using posterior segment OCT images.

In 2020, Chua et al. [11] trained a deep CNN to predict categories of refractive error using 305 smartphone-captured eccentric photorefraction images, achieving an overall accuracy of 81.6%. However, their participants were all of one homogenous ethnicity (Korean), whereas our study included patients of Indian, European, and Māori descent. Furthermore, their study comprised only patients aged six months to 8 years, while our cohort had a much greater age range of participants. Our study also had a larger sample size of 512 participants compared to 164. Images used in their study were taken with a 16-megapixel LG smartphone camera, and the other aforementioned smartphone-based photorefractive studies have all used distinct models, such as the Nokia Lumia 1020 [54], the iPhone 7 [30], and the OPPOA37f [53]. Unless the smartphone model and/or camera settings are somewhat standardized, the consistency of results may be affected. The nun IR camera used in our study has been specifically designed to be compatible with Android smartphones and comes together with a Samsung A03 pre-installed with the nun IR mobile app. The nun IR camera used in our study has inbuilt optics for imaging the eye and only uses smartphone as a user interface. Hence, the images acquired are not dependent on the camera specifications of the smartphone used. As nun IR is also a fundus camera, unlike regular smartphone cameras, it can complete a more comprehensive screening assessment, examining pupillary red reflexes and the retinal fundus.

Our study has several limitations. Firstly, a considerable number of photographs could not be included for reasons such as excessive brightness and low quality, thus rendering the pupillary crescents undetectable. This may have been due to poor technique in image acquisition, as this is user-dependent and could be improved by a more rigorous and meticulous data collection system. Secondly, our DL model produced better results for true positives compared to true negatives. As the nun IR camera was primarily designed to be a fundus camera, the optics could be adjusted to enhance the visibility of the pupillary crescents in IR mode if the camera were to be repurposed for myopia detection. Thirdly, we used autorefraction to measure refractive error, despite cycloplegic refraction being the gold standard [30]; thus, accommodation may have needed to be sufficiently controlled. Finally, although our study had a wide age range of participants, we did not include any children under 16 years. As many vision screening programs are focused on the early detection of refractive error, future studies should also encompass a greater pediatric cohort. Further, since the proposed study is based on comparatively fewer data samples, the generalizability of the model is limited.

Future research that could address the above limitations can definitely improve the overall performance of such red reflex image-based refractive error estimation systems.

## 5. Conclusions

In summary, we developed a DL-based model to estimate uncorrected refractive error using red reflex images captured by the nun IR fundus camera. The proposed approach achieved 75% accuracy in predicting spherical power with a multiclass classifier.

The results are inferior and do not achieve the target we aimed for as a primary screening tool. However, it demonstrates a definite potential for future research in vision screening with better data collection procedures in place, a larger sample size, better targeted CNN architectures, and additional collection of pediatric data.

## Figures and Tables

**Figure 1 diagnostics-13-02810-f001:**
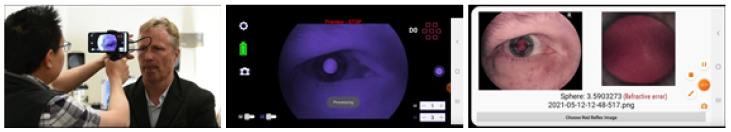
IR and colour red reflex image capture using nun IR.

**Figure 2 diagnostics-13-02810-f002:**
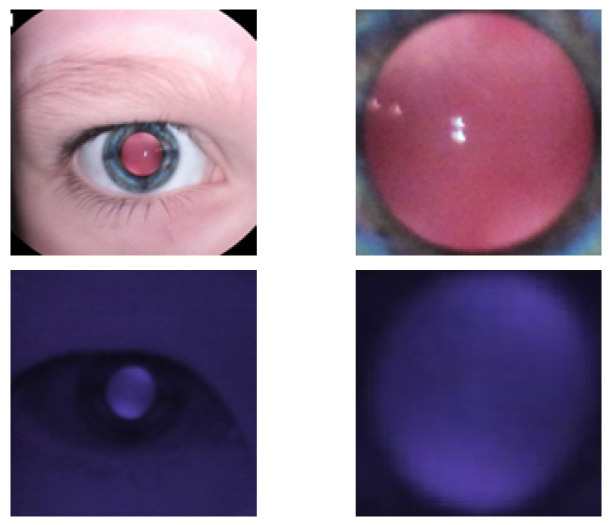
Images of pupils cropped from colour and IR images. Column 1 is the original image. Column 2 is cropped version.

**Figure 3 diagnostics-13-02810-f003:**
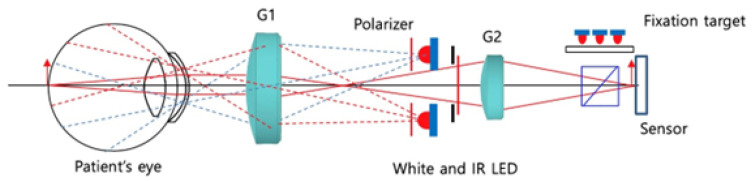
nun IR illumination path [50].

**Figure 4 diagnostics-13-02810-f004:**
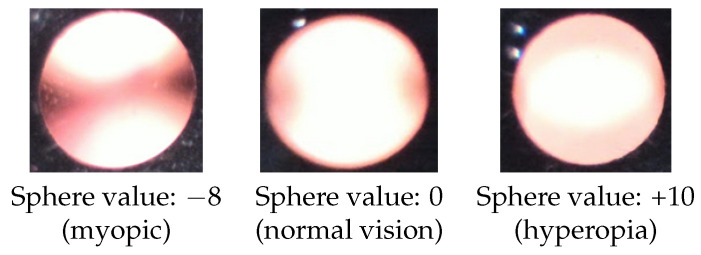
Red reflex color images taken by nun IR using a fake eye, for which spherical power can be adjusted.

**Figure 5 diagnostics-13-02810-f005:**
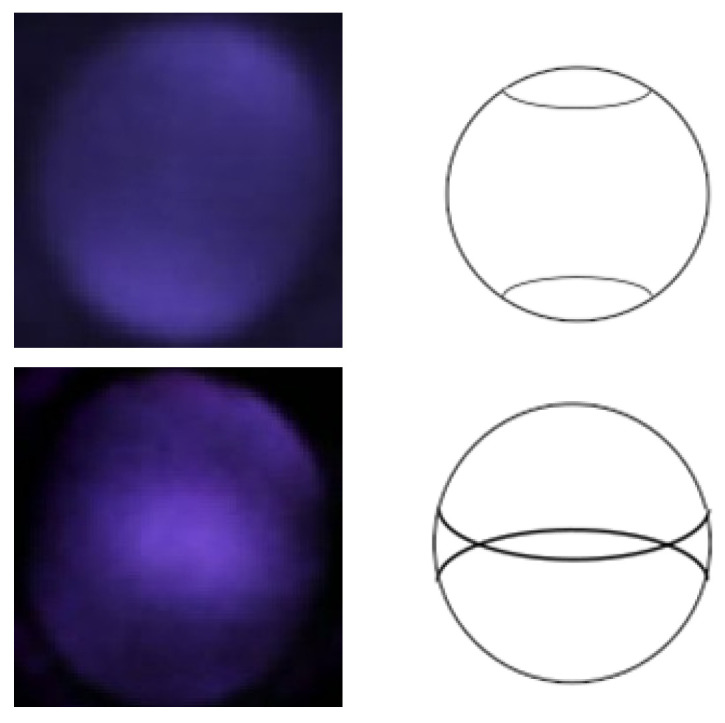
IR images with corresponding crescent types.

**Figure 6 diagnostics-13-02810-f006:**
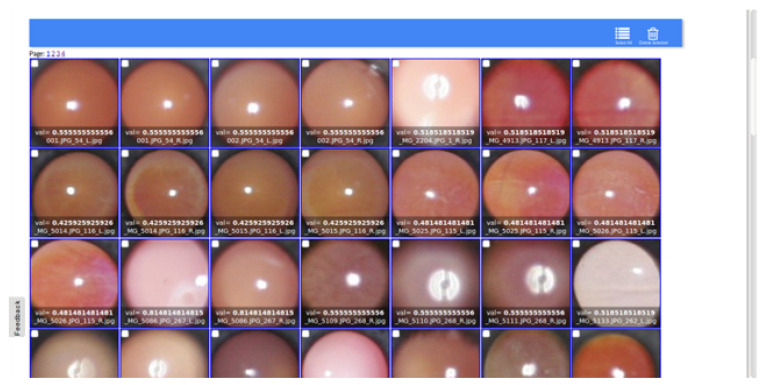
Pupils cropped automatically using MedicMind-AI portal.

**Figure 7 diagnostics-13-02810-f007:**
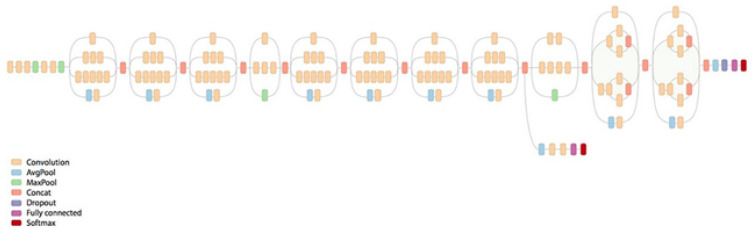
Inception-v3 architecture.

**Figure 8 diagnostics-13-02810-f008:**
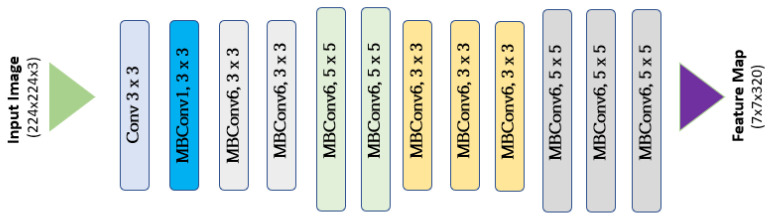
EfficientNet architecture.

**Figure 9 diagnostics-13-02810-f009:**
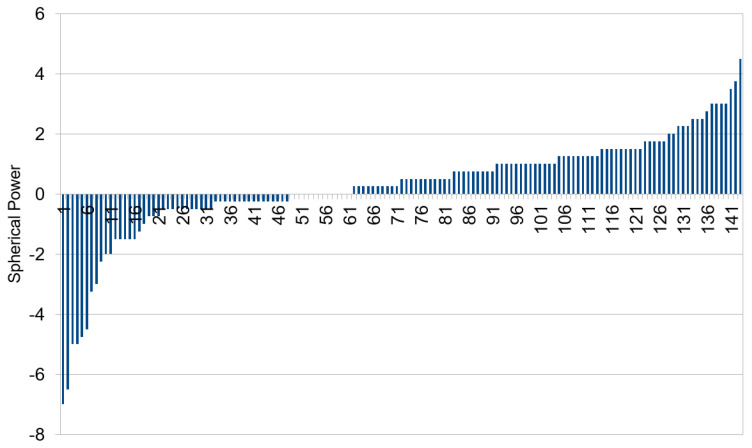
Spherical power distribution in the Dargaville dataset.

**Figure 10 diagnostics-13-02810-f010:**
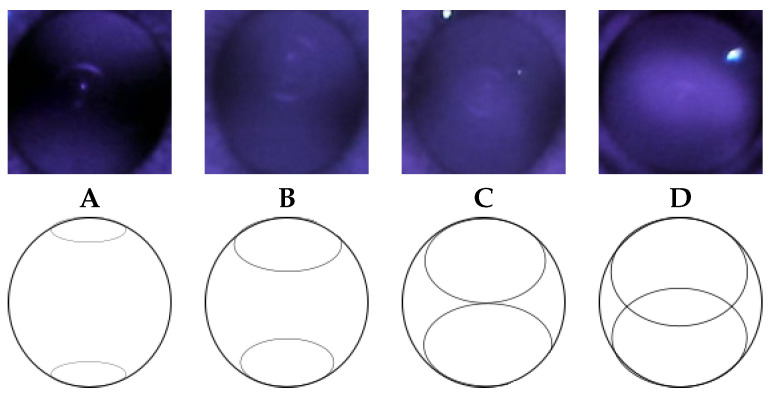
Crescent type categories (**A**–**D**).

**Figure 11 diagnostics-13-02810-f011:**
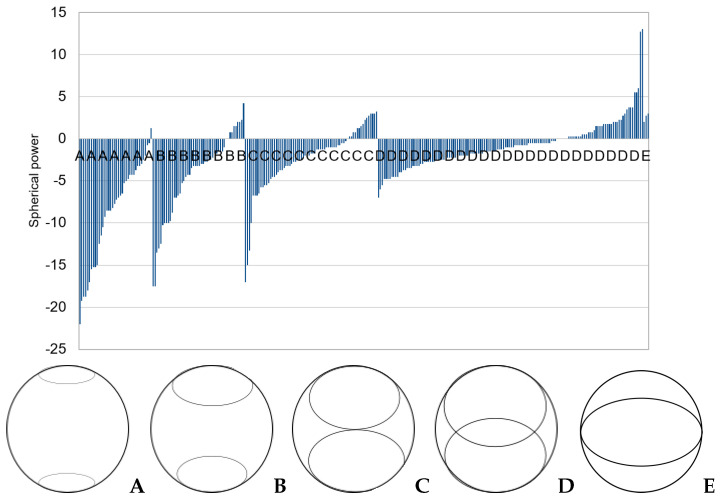
Sphere vs. crescent type in the Choithram dataset.

**Figure 12 diagnostics-13-02810-f012:**
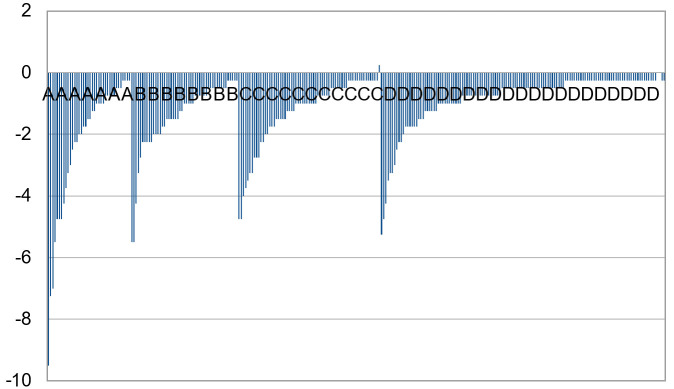
Cylinder vs. crescent type.

**Figure 13 diagnostics-13-02810-f013:**
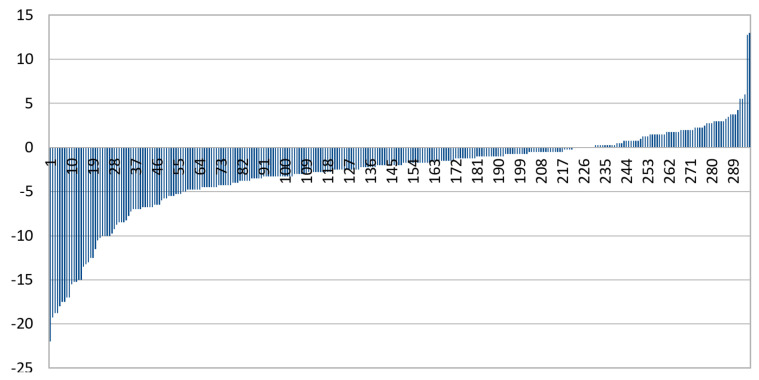
Distribution of spherical power for 288 IR red reflex images.

**Figure 14 diagnostics-13-02810-f014:**
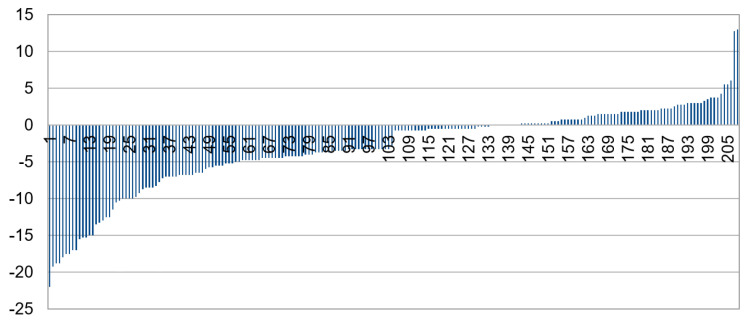
Distribution after removing median 30% of IR images.

**Figure 15 diagnostics-13-02810-f015:**
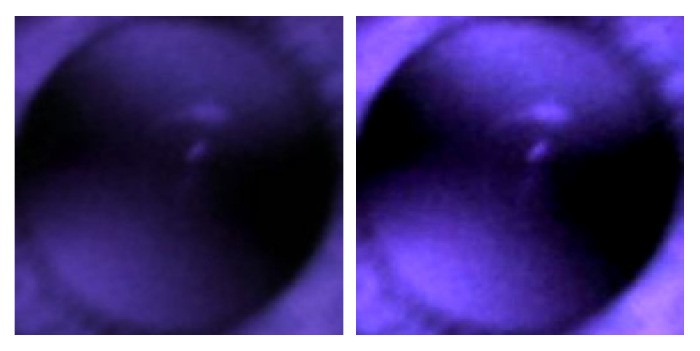
IR image before (**left**) and after (**right**) increasing contrast.

**Figure 16 diagnostics-13-02810-f016:**
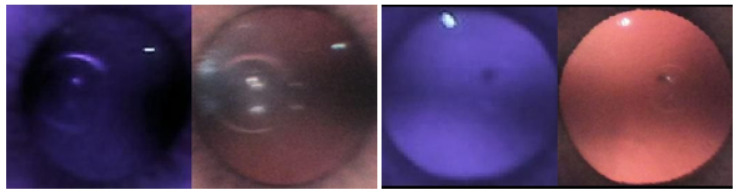
Combined color and IR images.

**Figure 17 diagnostics-13-02810-f017:**
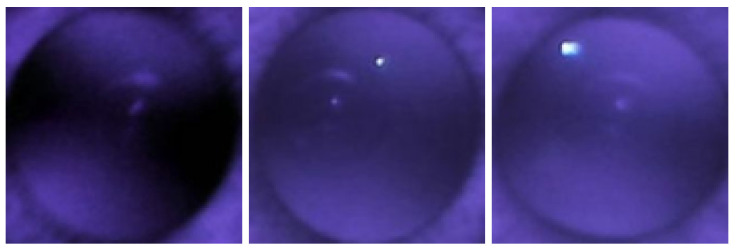
Myopic IR images.

**Figure 18 diagnostics-13-02810-f018:**
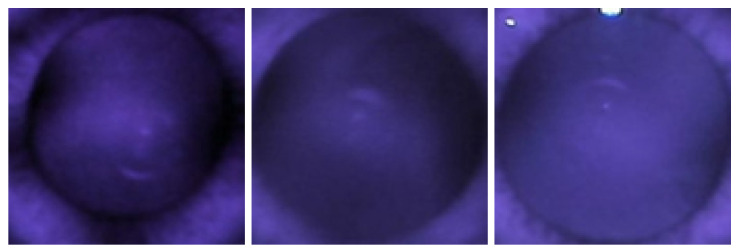
Normal (left image) and myopic (second and third image) IR images.

**Figure 19 diagnostics-13-02810-f019:**
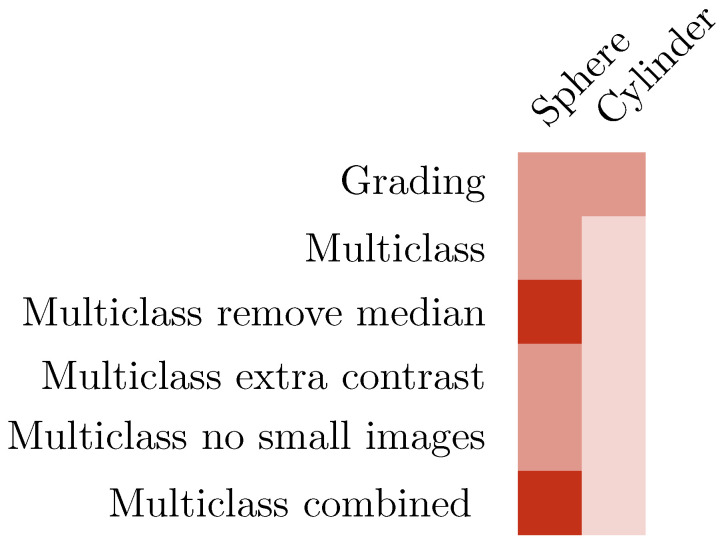
Accuracy of techniques for IR images using EfficientNet (≥70% dark, ≥60% medium otherwise light).

**Table 1 diagnostics-13-02810-t001:** Performance of different architectures.

	IR Inception-v3	IR EfficientNet	Colour Inception-v3	Colour EfficientNet
Sphere	55%	63%	54%	64%
Cylinder	49%	70%	50%	57%

**Table 2 diagnostics-13-02810-t002:** Performance of the multiclass classifier.

	IR Inception-v3	IR EfficientNet	Colour Inception-v3	Colour EfficientNet
Sphere	55%	70%	50%	64%
Cylinder	48%	55%	56%	60%

**Table 3 diagnostics-13-02810-t003:** Accuracy when removing median 20% of images.

	IR Inception-v3	IR EfficientNet	Colour Inception-v3	Colour EfficientNet
Sphere	75%	74%	63%	68%
Cylinder	49%	49%	49%	60%

**Table 4 diagnostics-13-02810-t004:** Effect of only training with red reflex images greater than 180 pixels in width for IR images and greater than 450 pixels in width for colour images.

	IR EfficientNet	Colour EfficientNet
20% spherical power median removed	74%	68%
Small-sized images and 20% spherical power median removed	64%	60%

**Table 5 diagnostics-13-02810-t005:** Accuracy when training with combined IR and color images.

	Inception-v3	EfficientNet
Sphere	74%	75%
Cylinder	60%	56%

## Data Availability

The datasets used and/or analyzed during the current study are available from the corresponding author upon reasonable request and upon the permission from the concerned authorities involved.

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
