# Peer review of "Automatic Refractive Error Estimation Using Deep Learning-Based Analysis of Red Reflex Images"

_diagnostics, 2023, doi:10.3390/diagnostics13172810_

Round 1
Reviewer 1 Report
I am pleased about the novelty of the study. However, as the authors have mentioned, the resulting accuracy of the model is concerning.
Few details to mention,
1) figure 4, 5 mentioned in page 6 seems to be erroneous.
2) 'resulting' should be 'Resulting mentioned in page 13 seems to be erroneous.
3) The reference number mentioning Chun et al. 6 seems to be erroneous.
Seems sound.
Author Response
Thanks for reviewing our article and providing valuable comments. It has helped us to improve the article.
We have addressed majority of the comments raised and have revised the article accordingly. A few of the comments, however, would require major changes to the research methodology that would be outside the initial research timeframe and funding. Hence, we were not able to address these in the revised version of the article, but will be definitely addressed through an immediate follow up research.
The detailed response to each comments can be found in the attached document.
Thanks again for reviewing our article and providing your valuable comments and suggestions.

Reviewer 2 Report
An interesting study with good use of statistics however there are many shortcomings some of which have been listed in the discussion. The sample is heterogeneous (2 different continents), the age group is adult (with much lesser accommodative effort), and the actual number of photographs being interpreted too less.
I would suggest that the author go through this work "Srivastava RM, Verma S, Gupta S, Kaur A, Awasthi S, Agrawal S*. Reliability of Smart Phone Photographs for School Eye Screening. Children. 2022; 9(10):1519.https://doi.org/10.3390/children9101519
The utility of any screening tool like the one being discussed is for children and more for hypermetropia than myopia for obvious reasons. Unfortunately, both these groups have not been studied. As has been discussed Autorefractor is not the gold standard (Cycloplegic refraction should have been performed). Only 20% of the photographs for validation is again a small number.
It would also be interesting to know how this screening tool compares with a simple screening method of taking vision on Snellen chart.
I also do not understand the rationale of trying to quantify the refractive error in screening . It would suffice to only detect the presence of refractive error by screening modality which can be quantified by an eye care personnel.
English is satisfactory.
Author Response

(The authors gave the same response as above.)

Round 2
Reviewer 2 Report
Acceptable
Satisfactory